# Psychometric Properties of the Condom Use Self-Efficacy Scale among Young Colombians

**DOI:** 10.3390/ijerph17113762

**Published:** 2020-05-26

**Authors:** Vanessa Sanchez-Mendoza, Encarnacion Soriano-Ayala, Pablo Vallejo-Medina

**Affiliations:** 1International PhD School, Universidad de Almeria, 04120 La Cañada, Spain; 2School of Psychology, Fundación Universitaria Konrad Lorenz, Bogotá 110231, Colombia; 3Department of Humanities and Education, Universidad de Almeria, 04120 La Cañada, Spain; esoriano@ual.es; 4SexLab KL—Human Sexuality Laboratory, School of Psychology, Fundación Universitaria Konrad Lorenz, Bogotá 110231, Colombia; pablo.vallejom@konradlorenz.edu.co

**Keywords:** HIV prevention, unintended pregnancies, condom use, sexual risk, Latins, psychometric, validity, sexual behavior, STI prevention

## Abstract

(1) Background: This study evaluated the psychometric properties of the Condom Use Self-Efficacy Scale among Colombian youth. (2) Method: A total of 2873 men and women between 18 and 26 years old (*M* = 21.45, *SD* = 2.26) took part in this study. All participants answered a socio-demographic survey, the Condom Use Self-Efficacy Scale, the UCLA Multidimensional Condom Attitudes Scale, The Condom Use Errors and Problems Scale, and the Sexual Assertiveness Scale. Sampling was web-based, and the survey was distributed via Facebook. (3) Results: The Condom Use Self-Efficacy Scale demonstrated adequate reliability (ordinal α ranged = 0.76 to 0.92). Exploratory and confirmatory factor analysis suggested a four-factor structure with an explained variance of 69%. This dimensionality was also invariant across gender. Moreover, positive attitudes toward condom use were significantly associated with appropriation and assertiveness. Two dimensions (appropriation and partner disapproval) showed significant gender differences. (4) Conclusions: The Spanish–Colombian version of the Condom Use Self-Efficacy Scale is a psychometrically adequate instrument to measure perceived condom use self-efficacy. This scale can be used in both research and professional settings to measure self-efficacy at using condoms in young people.

## 1. Introduction

Sexual health is a significant interest area for public health in Colombia. The Colombian government’s 2030 Agenda for Sustainable Development has established goals and indicators concerning health and well-being. Some of those goals are related to sexual and reproductive health: (1) to reduce the HIV/AIDS deaths tolls from 4.9 per 1000 people to 2.4 per 1000 people, and (2) to increase the percentage of women between the ages of 15 and 49 years old who use contraceptive methods from 68.1% to 81.4%. Those goals will be achieve over the next ten years [1]. Additionally, the government created the Ten-Year Public Health Plan 2012–2022 and The National Sexual and Reproductive Rights Policy. Both documents are written understanding health as a human right, gender differences in access to healthcare, taking a life course approach, and attending to social and behavioral factors [2]. 

In 2018, 36.2 million people lived with HIV/AIDS around the world, 1.7 million individuals became infected in 2018. Although the Caribbean, Central Africa, Europe, and North America reported significant decreases in new infections between 2010 and 2018, Latin America saw a 7% increase in cases [3]. In Colombia, HIV increased by even more than Latin America as a whole. Between 2015 and 2019, new HIV infections increased by 17.3%. The distribution of new infections by population group (among people aged 19–45 years) in Latin America shows that 40% of new infections occur amongst men who have sex with other men, 25% occur amongst sex workers, illegal drug users, and transgender people, and 35% occur in other segments of the population. In 2019, Colombia had 17,502 new cases. Males comprised 80.8%, and sexual intercourse was the cause of infection in most cases (17,219) [4]. 

Other sexually transmitted infections (STIs) are also a major problem. Global data show that more than a million people are infected daily, and an estimated 376 million new infections occur each year [5]. More than 500 million people are estimated to suffer from the herpes simplex virus, and more than 290 million women have one kind of human papillomavirus infection [6]. In Colombia, 94,000 people sought medical attention due to an STI between 2009 and 2011; according to individual service provision records, the age of highest prevalence was between 20 and 29 years of age, and the female population was the most affected [7]. The impacts of these infections vary across populations, depending on the biological, social, behavioral, and economic factors affecting also the reactions to and physiological results of STIs. Some of these are infertility [8], social inequality [5], depression and associated mental health disorders [9], and increases in healthcare expenditures [10].

Finally, another problem related to sexual health is unintended pregnancies. Latin America is one of the regions with the highest rate of unintended pregnancies; 122 out of 1000 women aged 15–44 are pregnant, but 55% were unintended. Latin America is the only region in the world where the percentage of unplanned pregnancies exceeds planned ones [11]. In Colombia, poverty and social factors are the major reasons to explain the high rates of teen pregnancy, one in every five women between 15 and 19 years of age are or have been pregnant and just 33.6% are planned [12]. Unintended pregnancies in women cause problems such as maternal death (830 women die during childbirth every day), it perpetuates the cycle of poverty in developing countries where 99% of mortality occurs and 7% of women drop out of school due to pregnancy, and finally it increases the risk of stillbirth due to preeclampsia, placental abruption, or maternal malnutrition [6].

Condoms are inexpensive and highly effective means at preventing HIV, STIs, and unintended pregnancies. In Colombia, consistent condom use is uncommon (22%) [13], its use oftentimes is doubtful and incorrect [14]. The greatest predictor of regular, consistent condom use is a person’s attitudes towards condoms (i.e., personal positive evaluation that condoms actually work) [15,16], subjective norms or beliefs (i.e., perceived normative support for condom use) [17,18,19] and self-efficacy (i.e., their belief that they can actually use condoms) [16,20]. Self-efficacy is supported by the social cognitive theory, according to which people’s behavior depends on how they evaluate their own abilities [21,22]. Thus, one of the reasons why young people take high risks when they engage in sexual activity may be associated with low perceived self-efficacy about their competence to negotiate condom use. Unlike the attitudes and subjective norms, there are no validated or translated standardized scales in Colombia to evaluate this key construct of sexual health.

Previous studies described above underscore the importance of socio-cognitive variables, such as self-efficacy, to increase condom use frequency and contribute to the prevention of STIs [23,24]. One of the scales used to assess self-efficacy at using condoms is the Condom Use Self-Efficacy Scale (CUSES) [25], which assesses a person’s feelings of confidence about buying condoms, putting them on and taking them off, and also the act of negotiating the use of it with a sexual partner. This scale has been used with college students from different cultural backgrounds [26] and other young people’s samples [23,24,27]. Validation studies have reported on the adequate psychometric properties of the CUSES as a valid and reliable instrument to measure condom use self-efficacy in different populations. Although developing a scale using the intended respondents’ native language remains the best option to measure self-efficacy adequately in a given context, validating and using an existing scale is not only economical, it also provides backgrounds for comparison in other settings [28].

The present study evaluated the psychometric properties of the CUSES in Colombia and established relationships between this scale and results from other validated instruments administered to Colombian participants. 

## 2. Method

### 2.1. Sample

Participants were recruited through Facebook. Initially, 3649 young people clicked on the scale; 281 of these acknowledged informed consent but failed to respond, seven failed to indicate their sex, 15 were over 26 years of age, 24 were not Colombian, and 449 did not answer the CUSES fully. A total of 2873 young people completed the sociodemographic questionnaire and the CUSES. Inclusion criteria were being between 18 and 26 years of age and being Colombian. Exclusion criteria were failure to provide informed consent or not completing the questionnaires. The mean age was 21.45 years (*SD* = 2.26). A total of 33.8% (*n* = 971) were male and 66.2% (*n* = 1902) were female. The characteristics of the sample are described in Table 1. 

Three subsamples were randomly obtained from the total sample; data from 873 participants were subjected to exploratory factor analysis (EFA), data from 1000 participants were subjected to confirmatory factor analysis (CFA), and data from 1000 participants were subjected to invariance analysis. Remaining analyses used the full sample.

### 2.2. Instruments

**Sociodemographic variables.** An ad hoc questionnaire was created in order to register sex orientation, age, nationality, and city of residence, schooling, gender, sexual orientation, and marital status of the participants. This questionnaire has previously been used in the Colombian context [13].

**Condom Use Self-Efficacy Scale (CUSES)** [25]. The scale evaluates a person’s feelings of confidence about buying condoms, putting them on and taking them off, and negotiating their use with a sexual partner. It is composed of 15 statements with five Likert-type response options from Strongly disagree = 0 to Strongly agree = 4; seven items are reverse-coded. Its original version includes four factors: (a) appropriation, with four items related to confidence in performing the technique necessary to use a condom in a sexual intercourse; (b) partner’s disapproval, including five items related to confidence in dealing with possible rejection from a sexual partner due to request to use a condom; (c) assertiveness, including three items related to an individual’s ability to ask a partner to use a condom; and (d) intoxication, with three items referred to the respondent’s confidence in their ability to use a condom while under the influence of alcohol or other drugs or when they are over-excited. The present study applied for first time the Colombian validated scale for Colombian youth [28] 29. Once reversed items (items 5 to 9) are re-coded, higher scores mean higher self-efficacy.

**The UCLA Multidimensional Condom Attitudes Scale (MCAS)** [29]. The MCAS scale evaluates people’s attitudes toward condom use. It is composed of five dimensions consisting of five items each (reliability, pleasure, stigma, negotiation, and shame). It is answered using Likert-type response options from 1 = totally disagree to 7 = totally agree. The present study applied the Colombian validated scale [30]. Higher scores indicate better attitudes toward the use of condoms. Obtained data in this study showed alphas ranging from 0.71 to 0.87.

**Condom Use Error/problems Survey (CUES)** [31]. The scale measures errors and problems associated with the use of condoms, considering the last three times a condom was used during the past three months as the recall period. Male (putting on a condom themselves) and female (putting the condom on their partner) versions are available. The scale is composed of 16 items with four Likert-type response options ranging from failure to perform an action to always performing it. High scores indicate a high frequency of errors/problems during any of the last three occasions or during the last three months. The present study applied the Colombian validated scale. [14].

**Sexual Assertiveness Scale (SAS)** [32]. The scale measures assertiveness with respect to initiation, refusal, and sexually transmitted disease/pregnancy prevention (STD-P P) with a regular partner. It is composed of 18 Likert-type items (Never = 0 to Always = 4). The brief (9 items) version was used, which has been validated to Colombia [33]. A high score represents high sexual assertiveness. The present study used only the initiation and STD-P P items. Initiation (assertiveness) is related to a person’s feelings of confidence about initiating sexual intercourse and making suggestions about physical intimacy. Alphas in the present study for those scales were 0.74 (initiation) and 0.85 (STD-P P).

### 2.3. Procedure

The questionnaire was published and distributed using Facebook. Payment of USD 150 was made to the virtual platform to promote the scale from 2 October to 17 October 2019. Responses were collected using a secure third-party survey provider (Survey Monkey, https://www.surveymonkey.com). The targeted population was young people aged 18–26. Online participants confirmed their willingness to participate through an online consent procedure, and questionnaire completion was also online. All internet protocol (IP) addresses were logged to discourage multiple responses from a single individual. By limited responses to one per IP address, the quality of the online data were higher.

**Data analysis.** Results were processed using R [34] (Version 3.6.0) and the R Studio interface [35] (Version 1.1.463). A polychoric matrix was for reliability and factorial analysis calculation; thus, the α presented is not Cronbach’s but ordinal. The number of dimensions to be extracted was calculated with the following methods: optimal coordinates, acceleration factor, parallel analysis, Eigenvalues (Kaiser criterion), Velicer Minimum Average Partial MAP, Bayesian Information Criterion BIC, sample size adjusted BIC, Very Simple Structure VSS complexity, and VSS complexity 2. The mode and the quality of the indicators indicated the *n* of factors. EFA was computed through an maximum likelihood robust ML-R) method using varimax rotation on the polychoric matrix of sub-sample 1. CFA was performed using a Weighted Least Square Mean and Variance Adjusted-Robust (WLSMV-R) estimator on a polychoric matrix based on sub-sample 2. Five different models were tested. The fit indexes used were root mean square error approximation (RMSEA) [36] and its 90% confidence interval (90% CI), the comparative fit index (CFI) [37], and Tucker Lewis Index (TLI) [38]. Values up to 0.08 for RMSEA are usually considered as acceptable, but it is desirable not to exceed a 0.06 threshold; while a value above 0.90 is acceptable, one higher than 0.95 is desirable for CFI and TLI [39]. For the invariance across gender, as before, data were derived from a polychoric matrix, in this case of sub-sample 3. WLSMV-R was also used. Invariance indicators were: a −0.01 change in CFI, paired with changes in RMSEA of +0.015 concerning the least restrictive model [40] and a non-significant increase of the χ^2^ using the Scaled Chi-Squared Difference Test [41]. Progressive invariance was tested for four models (configural, metric, scalar, and strict).

The following packages were also used: ggplot2 for data visualization [42] (Version 3.1.1), psych (Version 1.8.12) psychometric (Version 2.2) and psycho (Version 0.4.9.1) [43,44,45] were used for estimating some psychometric properties. While lavaan [46] (Version 0.6–5), semPlot [47] (Version 1.1.2), and semTools [48] were used for calculating and plotting the Structural Equation Model.

## 3. Results

The number of dimensions to isolate was evaluated using sub-sample 1. The methods (optimal coordinates, parallel analysis, Kaiser, and VSS complexity 2) suggested a three-dimensional model. Some of these methods are popular (Kaiser) and often recommended (optimal coordinates and parallel analysis). Nevertheless, we decided to compare the original four-dimensional structure of the scale with the three-dimensional structure (Table 2). The three-dimension (3-D) proposal explains 62% of the variance and has RSMEA = 0.127 and TLI = 0.85 as exploratory estimates. The four-dimension (4-D) model explains 69% of the variance, and the exploratory indicators RMSEA and TLI were 0.086 and 0.93, respectively. Item complexity was similar in both models; therefore, CFA results were needed to decide on the model.

CFA was performed using the second sub-sample. Five different models were tested: (1) a one-dimension model (1-D); (2) the three-dimension model previously explored by independent factors (3D I); (3) the three-dimension model previously explored with related factors (3D R); (4) four independent factors explored previously and proposed by previous theory (4D I); and finally, (5) four related factors explored previously and proposed by previous theory [25] (4D R). The main results are shown in Table 3. Model 4D R seems to have the best fit; it is the only one in which all fit indexes are acceptable. Standardized weights for the chosen model are shown in Figure 1.

Once the four-dimension model was confirmed to be the best, an analysis of gender invariance was performed. For that purpose, we used sub-sample 3. As shown in Table 4, a strict level of invariance was achieved. Based on the three indicators taken into account for the four levels, just a significant increase of the χ^2^ was observed for the weak invariance. However, the other indicators (ΔCFI and ΔRMSEA) seem to behave appropriately to suggest gender invariance.

The psychometric properties of the items began to be obtained. Once strict invariance across gender was tested and no differences were found, descriptive psychometric properties were assessed. In Table 5, ordinal alphas above 0.75 for all dimensions are shown. A couple of items (3 and 15) improve α if eliminated, but no further actions were considered necessary. Overall, corrected correlations between total and items are higher than 0.50. Item distributions cannot be considered normal, especially when considering kurtosis.

Criterion validity was also explored. To this aim, a correlation matrix between the CUSES subscales and other theoretically related instruments was created. Positive and significant correlation were observed and confirmed. Significantly low and moderate correlations were observed (see Table 6). Finally, the CUSES sub-scales were compared for gender. Figure 2 shows significant differences in appropriation and partner’s disapproval (see Figure 2A). Men (*M* = 13.86; *SD* = 2.68) have more appropriation of condom use than women (*M* = 11.85; *SD* = 3.45), and women’s score (*M* = 17.68; *SD* = 3.07) is higher in partner’s disapproval (women are better at handling disapproval) than men’s score (*M* = 16.9; *SD* = 3.45) (see Figure 2B). Neither assertiveness (men (*M* = 10.89; *SD* = 1.70); women (*M* = 10.82; *SD* = 1.77)) nor self-control (men (*M* = 8.77; *SD* = 2.47); women (*M* = 8.80; *SD* = 2.40)) show significant differences between men and women (see Figure 2C,D).

## 4. Discussion

This study assessed construct and criterion validity of the CUSES [25] adaptation into Colombian Spanish language, as well its reliability and certain psychometric properties. From the socio-cognitive perspective, studies on the reduction of sexual risk behaviors among young people are widespread and cover previously mentioned factors, such as knowledge about condom use, stigma related to its acquisition, and others. The use of CUSES in the present study contributes to the recognition of perceived self-efficacy as a factor mediated by aspects such as self-control, partner´s disapproval, and condom use appropriation.

Exploratory and confirmatory factor analyses suggested a four-factor model (appropriation, partner´s disapproval, assertiveness, and self-control). This four-factor structure was found to be invariant across gender. Reliability indexes and factors related with other similar variables were adequate. Gender-based differences were also observed. These results show that the current version of the CUSES is valid and reliable for its use in Colombia. 

Our first factorial analysis suggested a three-factor scale. Shaweno and Tekletsadik [24] also observed a three-dimensional structure in a brief 9-item version. However, the majority of previous studies [25,26,28] observed four dimensions. Therefore, we also considered a four-dimension structure with even better indicators. This model explained 69% of the variance. Other studies have shown a range of variance ranging from 48.2% [23] to 73.72% [28]. Furthermore, the confirmatory analysis presented better-fit indexes for this four-dimension model that was finally settled down with a strict invariance across gender. To the best of our knowledge, this is the first CUSES study including a strict invariance test. This level of invariance allowed for the comparison of the mean of the factors with a minimal bias across gender due to variance and covariance errors [49]. In other words, items were measured with the same precision in each group and no gender biases were found, this factor is key to better understand sexual behaviors [50]. Our fourth factor was termed self-control, which is consistent with the “intoxicants” [26], “pleasure and intoxicants”, [28] or “intoxicant control” [24] factors. The different label used in our study was due to the fact that, in the cultural adaptation, we used an adjustment of cultural words and context [51]. Thus, “heat of passion” is not related to the consumption of alcohol or psychoactive substances in the Colombian cultural context, although, taken together, all these factors are associated with self-control. 

All items show adequate psychometric properties and total item correlations above 0.50, as recommended by the literature [52]. In addition, slight improvements can be observed when removing item 3 of the appropriation factor (“I feel confident I could gracefully remove and dispose of a condom when we have intercourse”) or item 15 of the self-control factor (“I feel confident I could stop to put a condom on myself or my partner even in the heat of passion”). However, its level of adjustment is good and the improvements if the items were eliminated are considered negligible. Another indicator is item improvements. Item distribution is not normal and a remarkably high kurtosis is found; however, standard deviations are close to 1, indicating adequate score variability [53]. Alphas are observed to be similar to those reported in other studies, with values ranging from 0.76 to 0.92. These values are considered adequate for both research and professional practice. 

Consistent with prior research, we find that the CUSES is related with attitudes toward condom use [27]. Concerning relationships between CUSES and other measures (i.e., criterion validity), they are found to be low or moderate in most cases. We find associations with the UCLA Multidimensional Condom Attitudes Scale (MCAS) factors, in which higher scores represent better attitudes toward condom use. Ritchwood, Penn, Peasant, Albritton, and Corbie-Smith [54] observed that greater condom use self-efficacy was predicted by favorable attitudes toward condom use, which is similar to the association between partner´s disapproval, and condom use negotiation observed in the present study. Likewise, the present study finds an association between partner´s disapproval and attitudes toward condom use stigma, which has been related with reduced safer sex practices [55]. These findings highlight the necessity of interventions or training programs to improve young people’s abilities to cope with factors that could result in not using a condom. Although the Colombian Ministry of Health has designed advertising campaigns and prevention programs aimed to young people population, it is important to design strategies considering academic curricula. Additionally, a low and inverse association between self-efficacy factors and Condom Use Errors and Problems (CUEP) factors is found: scores on condom use self-efficacy increase as those associated with errors and problems with condom use decrease, which has previously been reported [56]. This observation is interesting because actual problems and errors seem to differ from one’s perception of the correct use of condoms, or self-efficacy. Significant relationship with variables such as initiation assertiveness (Init) and sexually transmitted disease/unintended pregnancy prevention assertiveness (STD-P P) were also verified according to assumptions. Thus, condom use self-efficacy certainly has an impact in delaying young people’s sex initiation and reducing sexually transmitted infections and unplanned pregnancies [57].

Gender comparisons show that men scored higher in appropriation and lower in partner´s disapproval than women. This may suggest a need for sexual health intervention programs aimed at increasing condom use self-efficacy differentially for men and women. For men, programs should emphasize how to deal with rejection by the sexual partner, and for women, training should be oriented toward appropriation. Differentiated interventions should also be designed to prevent STIs and unintended pregnancies. In this regard, the instrument’s subscales provide information on the effectiveness of negotiating condom use that could be associated with cultural factors in the Colombian context [58]. Highly typified gender roles, benevolent sexism, and prejudices associated with female sexuality and women’s sexual autonomy [59,60] have also been described by Peasant, Parra, and Okwumabua [61] as factors affecting condom use self-efficacy among women, including their ability to successfully negotiate condom use without compromising their relational goals.

A relevant strength of the present study is the sample size and its heterogeneity across the cities, that means that the CUSES can be used to obtain a reliable and valid measurement of condom use self-efficacy among Colombian young people, and it can be used as a baseline characterization scale for other Latin American Spanish-speaking populations to help in the early identification of youth condom use self-efficacy. Finally, this is the first successful adaptation in the continent.

Limitations of the present study is the non-randomized sampling approach used, the sample was skewed female and heterosexual an, the study excluded people without internet access because of the web-based survey method used. Future studies can further explore the differences in scores obtained by Spanish speaking groups at high risk of acquiring HIV/STI (an example, homosexual males, men who have sex with men and sex workers).

## 5. Conclusions

The CUSES is the first instrument used in Colombia to measure condom use self-efficacy in Spanish language. The reliability of the CUSES has been established in different studies since 1994 [23,24,25,26,27]. The present study allows CUSES use in Colombia with enough psychometric guarantees for its use for both, social/health and research interest. Some gender differences in condom self-efficacy and uncommon condom use were observed, suggesting the need to implement sexual health programs in Colombia considering these differences.

## Figures and Tables

**Figure 1 ijerph-17-03762-f001:**
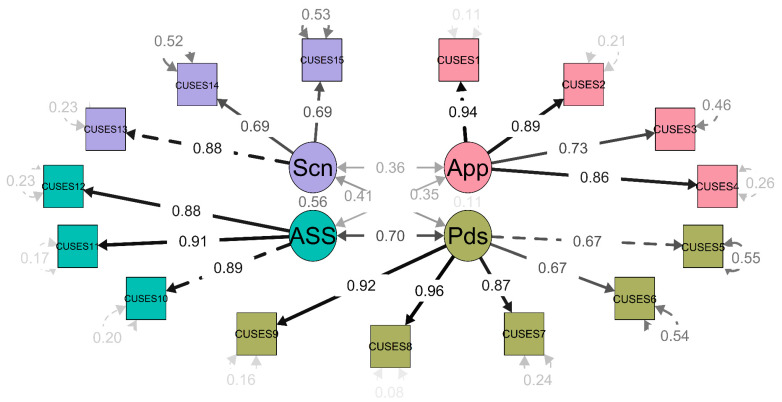
Confirmatory factor analysis (CFA) path diagram on four proposed dimensions. Standardized weights are presented. *App* = appropriation; *Pds* = partner’s disapproval; *Ass* = assertiveness; *Scn* = self-control.

**Figure 2 ijerph-17-03762-f002:**
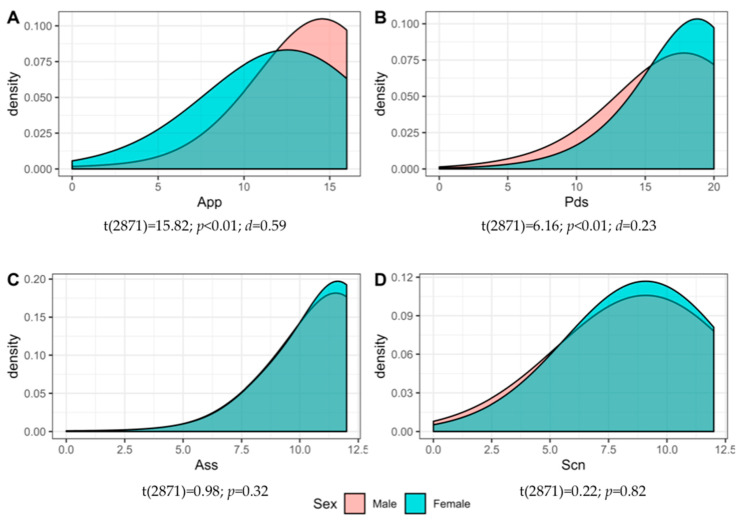
Densiogram distribution across gender. (**A**) gender differences in appropriation (App); (**B**) gender differences in partner’s disapproval (Pds); (**C**) gender differences in assertiveness (Ass); (**D**) gender differences in self-control (Scn).

**Table 1 ijerph-17-03762-t001:** Sample description.

	Males (*n* = 971)	Females (*n* = 1902)	Total (*n* = 2873)
	*N* %	*N* %	*N* %
*Age*			
18	81 (8.3)	230 (12.1)	311 (10.8)
19	109 (11.2)	267 (14)	376 (13.1)
20	134 (13.8)	282 (14.8)	416 (14.5)
21	131 (13.5)	287 (15.1)	418 (15.5)
22	123 (12.7)	236 (12.4)	359 (12.5)
23	127 (13.1)	239 (12.6)	366 (12.7)
24	128 (13.2)	154 (8.1)	282 (9.8)
25	107 (11)	163 (8.6)	270 (9.6)
26	31 (3.2)	44 (2.3)	75 (2.6)
*Schooling level N (%)*			
Basic Elementary	1 (0.1)		1 (0.0)
Secondary	52 (5.4)	101 (5.3)	153 (5.3)
First Technical level	59 (6.1)	142 (7.5)	201 (7)
Second Technical level	51 (5.3)	85 (4.5)	136 (4.7)
College undergraduate	573 (59)	1159 (60.9)	1732 (60.3)
College graduate	183 (18.8)	323 (17)	506 (17.6)
Postgraduate candidate	41 (4.2)	48 (2.5)	89 (3.1)
Postgraduate	11 (1.1)	44 (2.3)	55 (1.9)
*Marital Status N (%)*			
Married	14 (1.4)	26 (1.4)	40 (1.4)
Single	867 (89.5)	1659 (87.5)	2526 (88.1)
Widowed	1 (0.1)		1 (0.0)
Co-habiting	81 (8.4)	206 (10.9)	287 (10)
Separated/Divorced	6 (0.6)	6 (0.3)	12 (0.4)
*Sexual orientation N (%)*			
Exclusively heterosexual	738 (76.2)	1431 (75.4)	2169 (75.6)
Predominantly heterosexual, only incidentally homosexual	66 (6.8)	297 (15.6)	363 (12.7)
Predominantly heterosexual, but more than incidentally homosexual	10 (1)	60 (3.2)	70 (2.4)
Equally heterosexual and homosexual	15 (1.5)	66 (3.5)	81 (2.8)
Predominantly homosexual, but more than incidentally heterosexual	10 (1)	9 (0.5)	19 (0.7)
Predominantly homosexual, only incidentally heterosexual	28 (2.9)	12 (0.6)	40 (1.4)
Exclusively homosexual	98 (10.1)	13 (0.7)	111 (3.9)
Asexual	4 (0.4)	11 (0.6)	15 (0.5)
*Do you have a partner that you have been seeing for more than 6 months?*			
Yes	491 (50.7)	1179 (62.1)	1670 (58.1)
No	477 (49.3)	719 (37.9)	1196 (41.6)
*How often do you use condoms during sexual intercourse?*			
Every time	271 (28)	334 (17.6)	605 (21.1)
Usually	248 (22.6)	345 (18.2)	593 (20.7)
Frequently	105 (10.8)	184 (9.7)	289 (10.1)
Sometimes	55 (5.7)	125 (6.6)	180 (6.3)
Occasionally	92 (9.5)	253 (13.3)	345 (12)
Rarely	105 (10.8)	321 (16.9)	426 (14.9)
Never	93 (9.6)	336 (17.7)	429 (15)

**Table 2 ijerph-17-03762-t002:** Exploratory factor analysis based on the polychoric matrix using maximum likelihood and varimax rotation.

	3 Dimensions						4 Dimensions							
	D3	D2	D1	h^2^	u^2^	com		D2	D3	D1	D4	h^2^	u^2^	com
CUSES8	**0.92**			0.89	0.10	10.1	CUSES8	**0.93**				0.89	0.11	1.1
CUSES9	**0.90**			0.88	0.12	10.2	CUSES9	**0.91**				0.88	0.11	1.1
CUSES7	**0.84**			0.78	0.21	10.2	CUSES7	**0.85**				0.78	0.21	1.1
CUSES5	**0.42**			0.26	0.73	10.8	CUSES5	**0.44**				0.26	0.74	1.7
CUSES6	**0.41**		0.32	0.27	0.72	10.9	CUSES6	**0.43**				0.27	0.73	1.9
CUSES1		**0.93**		0.91	0.09	10.1	CUSES1		**0.94**			0.91	0.09	1.1
CUSES2		**0.89**		0.85	0.15	10.2	CUSES2		**0.89**			0.85	0.15	1.1
CUSES4		**0.85**		0.78	0.22	10.1	CUSES4		**0.86**			0.78	0.22	1.1
CUSES3		**0.76**		0.65	0.35	10.2	CUSES3		**0.77**			0.65	0.35	1.2
CUSES11	0.40		**0.80**	0.85	0.15	10.7	CUSES13			**0.96**		0.98	0.02	1.1
CUSES10	0.35		**0.77**	0.77	0.23	10.6	CUSES14			**0.66**		0.47	0.53	1.1
CUSES12	0.39		**0.71**	0.70	0.30	10.8	CUSES15			**0.43**		0.31	0.68	2.5
CUSES13			**0.48**	0.28	0.72	10.4	CUSES11	0.44			**0.75**	0.88	0.11	2.2
CUSES15			**0.41**	0.23	0.76	10.8	CUSES10	0.40			**0.71**	0.77	0.22	2.1
CUSES14			**0.35**	0.15	0.85	10.4	CUSES12	0.43			**0.60**	0.69	0.31	2.7
Variance	0.22	0.21	0.19			*M* = 10.4	Variance	0.23	0.22	0.12	0.12			*M* = 1.5 *

* *D =* Number of dimensions’ model; *h^2^* = communality of the item; *u^2^=* uniqueness of the item; *com*= Hoffmann’s item complexity. Weights lower than 0.30 are hidden; boldface represent correct item-factor weight.

**Table 3 ijerph-17-03762-t003:** Fit indexes for the model tested.

Models	χ^2^	df	*p*	CFI	TLI	RMSEA	90% CI RMSEA
D-1	4284.53	90	<0.01	0.81	0.78	0.216	0.211–0.222
D-3 I	4146.70	90	<0.01	0.82	0.79	0.212	0.207–0.218
D-3 R	920.65	87	<0.01	0.96	0.95	0.098	0.092–0.104
D-4 I	4242.12	90	<0.01	0.81	0.78	0.215	0.209–0.220
D-4 R	432.33	84	<0.01	0.98	0.98	0.064	0.058–0.071 *

* *χ^2^ =* Chi-Square statistic; *Df =* degrees of freedom; *CFI =* Comparative Fit Index; *TLI =* Tucker Lewis Index; *RMSEA =* root mean square error of approximation.

**Table 4 ijerph-17-03762-t004:** Fit indexes for the invariance 4-D R model.

Models	χ^2^	*p* > χ^2^	df	*p*	CFI	ΔCFI	RMSEA	ΔRMSEA
Conf	295.57	-	168	<0.01	0.939	-	0.039	-
Metric	322.49	0.012 *	179	<0.01	0.932	−0.007	0.040	0.001
Scalar	332.95	0.561	190	<0.01	0.932	0	0.039	−0.001
Strict	354.49	0.054	205	<0.01	0.929	−0.003	0.038	−0.001 *

* *χ^2^ =* Chi-square statistic; *Df =* degrees of freedom; *Conf =* configural invariance; *Metric =* metric invariance; *Scalar =* strong invariance; *Strict =* strict invariance.

**Table 5 ijerph-17-03762-t005:** Some psychometric item properties.

Dim	Item	*M*	*SD*	Skew	Kurtosis	Ci-tc	α-Item	α
App	1	3.18	0.99	−1.23	1.08	0.88	0.88	0.92
	2	3.31	0.88	−1.48	2.30	0.86	0.89	
	3	3.25	0.92	−1.30	1.54	0.73	0.93	
	4	2.80	1.09	−0.68	−0.30	0.82	0.90	
Pds	5	3.40	0.94	−1.75	2.60	0.61	0.89	0.89
	6	3.25	0.97	−1.30	1.09	0.59	0.89	
	7	3.58	0.84	−2.49	6.24	0.82	0.84	
	8	3.62	0.77	−2.57	7.13	0.83	0.84	
	9	3.57	0.84	−2.30	5.14	0.81	0.84	
Ass	10	3.60	0.69	−2.18	5.95	0.84	0.88	0.92
	11	3.67	0.60	−2.25	6.87	0.87	0.85	
	12	3.58	0.71	−2.11	5.47	0.80	0.91	
Scn	13	2.99	0.96	−0.70	−0.01	0.89	0.51	0.76 *
	14	2.71	1.13	−0.55	−0.34	0.82	0.69	
	15	3.09	0.99	−0.96	0.35	0.76	0.81	

* *M* = mean; *SD* = standard deviation; *CI‒TC* = corrected item-total correlation; *App* = appropriation; *Pds* = partner’s disapproval; *Ass* = assertiveness; *Scn* = self-control.

**Table 6 ijerph-17-03762-t006:** Means, standard deviations, and correlations with confidence intervals.

Variable	*M*	*SD*	1	2	3	4	5	6	7	8	9	10	11
1. App	12.53	3.35											
2. Pds	17.42	3.23	0.08 **										
			[0.04, 0.12]										
3. Ass	10.85	1.75	0.27 **	0.48 **									
			[0.24, 0.31]	[0.45, 0.51]									
4. Scn	8.79	2.43	0.27 **	0.24 **	0.36 **								
			[0.24, 0.31]	[0.20, 0.27]	[0.33, 0.40]								
5. Neg	9.18	4.56	0.25 **	0.54 **	0.54 **	0.30 **							
			[0.28, 0.21]	[0.57, 0.52]	[0.57, 0.52]	[0.33, 0.26]							
6. Reli	11.87	5.11	0.20 **	0.11 **	0.15 **	0.14 **	0.20 **						
			[0.23, 0.16]	[0.14, 0.07]	[0.19, 0.11]	[0.18, 0.10]	[0.16, 0.24]						
7. Plea	16.85	6.08	0.14 **	0.23 **	0.20 **	0.29 **	0.25 **	0.21 **					
			[0.17, 0.10]	[0.26, 0.19]	[0.24, 0.16]	[0.32, 0.25]	[0.22, 0.29]	[0.17, 0.24]					
8. Sham	13.72	7.32	0.28 **	0.19 **	0.26 **	0.21 **	0.32 **	0.15 **	0.17 **				
			[0.32, 0.25]	[0.23, 0.15]	[0.29, 0.22]	[0.25, 0.17]	[0.29, 0.36]	[0.11, 0.18]	[0.13, 0.20]				
9. Stig	7.43	3.30	0.07 **	0.48 **	0.38 **	0.20 **	0.48 **	0.15 **	0.26 **	0.21 **			
			[0.11, 0.03]	[0.51, 0.45]	[0.41, 0.34]	[0.24, 0.17]	[0.45, 0.51]	[0.12, 0.19]	[0.23, 0.30]	[0.17, 0.25]			
10. CUEP	9.67	4.46	−0.15 **	−0.19 **	−0.11 **	−0.16 **	0.24 **	0.15 **	0.44 **	0.12 **	0.18 **		
			[−0.21, −0.09]	[−0.25, −0.13]	[−0.17, −0.04]	[−0.22, −0.10]	[0.18, 0.30]	[0.09, 0.21]	[0.39, 0.49]	[0.06, 0.19]	[0.12, 0.24]		
11. Init	7.86	3.13	0.15 **	0.19 **	0.19 **	0.11 **	−0.23 **	−0.07 **	−0.07 **	−0.15 **	−0.17 **	0.04	
			[0.12, 0.19]	[0.15, 0.23]	[0.15, 0.22]	[0.07, 0.14]	[−0.26, −0.19]	[−0.11, −0.03]	[−0.10, −0.03]	[−0.19, −0.11]	[−0.21, −0.13]	[−0.03, 0.10]	
12. STI−P	6.37	4.11	0.06 **	0.22 **	0.20 **	0.24 **	−0.24 **	−0.16 **	−0.39 **	−0.04	−0.18 **	−0.32 **	−0.02 **
			[0.02, 0.10]	[0.18, 0.26]	[0.17, 0.24]	[0.20, 0.28]	[−0.28, −0.21]	[−0.20, −0.12]	[−0.42, −0.35]	[−0.08, 0.00]	[−0.21, −0.14]	[−0.37, −0.26]	[−0.06, 0.02]

*M* and *SD* are used to represent mean and standard deviation, respectively. Values in square brackets indicate the 95% confidence interval for each correlation. The confidence interval is a plausible range of population correlations that could have caused the sample correlation (Cumming, 2014). ** indicates *p* < 0.01. *App* = appropriation CUSES; *Pds* = partner’s disapproval CUSES; *Ass* = assertiveness CUSES; *Scn* = self-control CUSES; *Neg* = embarrassment about negotiation and use of condoms UCLA; *Reli* = reliability and effectiveness of condoms UCLA; *Plea* = sexual pleasure associated with condom use UCLA; *Sham* = embarrassment about the purchase of condoms UCLA; *Stig* = stigma attached to persons who use condoms UCLA; *CUEP* = condom use errors/problems; *Init* = initiation Sexual Assertiveness Scale (SAS); *STI_P_P* = STI and pregnancy prevention SAS.

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
