# Peer review of "Psychometric Properties of the Condom Use Self-Efficacy Scale among Young Colombians"

_ijerph, 2020, doi:10.3390/ijerph17113762_

Round 1
Reviewer 1 Report
Dear authors,
Let me start off my saying that I really admire the work of scientists who are not native English speakers writing for an English-speaking audience. We are lucky to see your work. Unfortunately, this manuscript needs some work on the clarity of the English text and I don't believe it can be accepted until the English is clearer.
A few examples:
1) The first sentence is awkward and needs either a "the" before the phrase "good health and well-being" or should more likely be restructured to say "the government established goals and indicators in the field of "good health and well-being." Though I'm also not sure why is needs to be in quotes.
2) The next sentence sounds a bit like it has been translated from Spanish. A clearer way to write this sentence would be "The Colombian government has two goals related to sexual and reproductive health: 1) to reduce the number of deaths from HIV/AIDS from 4.9 per 1,000 people to 2.4 per 1,000 people and 2) to increase the percent of women between the ages of 15 and 49 who use contraception from 68.1% to 81.4%. Both of these goals are 10 year goals." The way you've written the sentence now is too confusing to be the second sentence in the paper.
3) The first sentence in the second paragraph should have "in 2018" or "globally" at the end. At the beginning it is disruptive to the flow. Also the numbers reported are confusing. For example, are the 1.7 million people INCLUDED in the 36.2 million or are they in addition? I think it would be clearer to say "In 2018, there were 36.2 million people living with HIV/AIDS around the world and of these people, 1.7 million were newly infected in 2018."
4) another example of an English challenge is that in numbers, the style for numbers is to use a comma instead of a period. So at the top of page 2, the 17,502 cases should use a comma and not a period (17.502 means 17 and a half).
Note that the English in the methods section is MUCH clearer. I wonder if someone else wrote this section? If so, having that person read the beginning and the end probably would make sense.
I think the methods are clear and sufficient but given the need to work on the front sections before this can be published, I didn't review this AS carefully.
In the discussion section, I think you need to highlight why you think it's important to test this in another setting. I think it definitely IS important but the discussion (or the intro) seems to be lacking the theoretical explanation for why testing in another setting is necessary and why we wouldn't expect the results to just be the same as elsewhere. This doesn't have to be very involved but I think you need to include it to justify the needed research.
Author Response
Dear reviewer
We are pleased to submit our revised manuscript entitled "Psychometric Properties of the Condom Use Self-Efficacy Scale among Young Colombians" for consideration by International Journal of Environmental Research and Public Health within its Special Issue "Policies and Strategies in Sexual and Reproductive Health"
We would like to thank you for giving us the opportunity to improve our manuscript as well as for your comments and recommendations. We have considered all of your suggestions and have incorporated them into the manuscript. We have responded point by point to your comments. We have track changes (red and green text) to record each change within the document.
Thank you for your consideration of this manuscript. Our responses are presented below.
Reviewer comment The first sentence is awkward and needs either a "the" before the phrase "good health and well-being" or should more likely be restructured to say "the government established goals and indicators in the field of "good health and well-being." Though I'm also not sure why is needs to be in quotes.
We agree this manuscript has been now review by a professional English editing service, it can be seen in "track changes".
Reviewer comment. The next sentence sounds a bit like it has been translated from Spanish. A clearer way to write this sentence would be "The Colombian government has two goals related to sexual and reproductive health: 1) to reduce the number of deaths from HIV/AIDS from 4.9 per 1,000 people to 2.4 per 1,000 people and 2) to increase the percent of women between the ages of 15 and 49 who use contraception from 68.1% to 81.4%. Both of these goals are 10 year goals." The way you've written the sentence now is too confusing to be the second sentence in the paper.
We have improved English and grammar including these examples.
Reviewer comment. The first sentence in the second paragraph should have "in 2018" or "globally" at the end. At the beginning it is disruptive to the flow. Also the numbers reported are confusing. For example, are the 1.7 million people INCLUDED in the 36.2 million or are they in addition? I think it would be clearer to say "In 2018, there were 36.2 million people living with HIV/AIDS around the world and of these people, 1.7 million were newly infected in 2018."
We have changed it as suggested.
Reviewer comment. Another example of an English challenge is that in numbers, the style for numbers is to use a comma instead of a period. So at the top of page 2, the 17,502 cases should use a comma and not a period (17.502 means 17 and a half).
Sure, done.
Reviewer comment. Note that the English in the methods section is MUCH clearer. I wonder if someone else wrote this section? If so, having that person read the beginning and the end probably would make sense.
Not really, maybe we make it simple. We do not know, maybe we had clearer ideas.
Reviewer comment. I think the methods are clear and sufficient but given the need to work on the front sections before this can be published, I didn't review this AS carefully.
The whole document has been reviewed, so we hope catching any mistake as well.
Reviewer comment In the discussion section, I think you need to highlight why you think it's important to test this in another setting. I think it definitely IS important but the discussion (or the intro) seems to be lacking the theoretical explanation for why testing in another setting is necessary and why we wouldn't expect the results to just be the same as elsewhere. This doesn't have to be very involved but I think you need to include it to justify the needed research.
We agree it has been included from line 116 onward and 630 onward.
We believe that the manuscript is now according to your considerations.
Sincerely,
Vanessa Sanchez-Mendoza
Researcher
Reviewer 2 Report
- Why do you use CUSES in the study? How importance of CUSES? It needs to explain in Introduction section in advance.
- What is the relationship between CUSES and prevention of STI? Please describe more detail in Introduction section.
- How to include the study subjects? Please describe the source of subjects more clearly in method.
- Add limitation in discussion section.
- Re-write the conclusion. Remove the limitation from conclusion.
Author Response
Dear reviewer
We are pleased to submit our revised manuscript entitled "Psychometric Properties of the Condom Use Self-Efficacy Scale among Young Colombians" for consideration by International Journal of Environmental Research and Public Health within its Special Issue "Policies and Strategies in Sexual and Reproductive Health"
We would like to thank you for giving us the opportunity to improve our manuscript as well as for your comments and recommendations. We have considered all of your suggestions and have incorporated them into the manuscript. We have responded point by point to your comments. We have track changes (red and green text) to record each change within the document.
Thank you for your consideration of this manuscript. Our responses are presented below.
Reviewer comment: Why do you use CUSES in the study?
It has been described from line 116 onward.
Reviewer comment: What is the relation between CUSES and STI
It has been describe from line 116 onward.
Reviewer comment: How to include the study subjects?
It has been included from line 253 onward.
Reviewer comment: Add limitations in discussion section
It has been included from line 774 onward.
Reviewer comment: Re-write the conclusion. Remove the limitations
It has been describe from line 779 onward.
We believe that the manuscript is now according to your considerations.
Sincerely,
Vanessa Sanchez-Mendoza
Round 2
Reviewer 1 Report
Dear authors,
This English writing in this is much improved since the first time I read this although there are still a number of issues. I've put extensive comments into the PDF which attached below that highlight specific sentences that are unclear, points where you don't share enough information, and some further limitations. I think this paper unfortunately still needs a fair amount of work before it's ready for publication. One thing to be aware of is that you need to guide the reader along a bit more carefully so that you are really walking them through your methods, figures, and conclusions. You present all of the findings a bit too methodically without, for example, even mentioning what Figure 2 really shows in the text. The text should help a reader interpret figures and right now you don't do enough of that.
You also still have several typos throughout - misspellings, extraneous commas, semicolons that don't make much sense. I think this needs a copy edit but also some careful additional clarity. I hope my comments are helpful.

Author Response
Dear reviewer
Special Issue "Policies and Strategies in Sexual and Reproductive Health"
International Journal of Environmental Research and Public Health
We would like to thank you for giving us the opportunity to improve our manuscript as well as for your comments and recommendations.
We have considered all of your extensive comments and have incorporated them into the manuscript. We have responded point by point to your PDF's comments.
1. We have improved English and grammar
2. We have clarified the highlight specific sentences
3. We have made copy edit
We believe that the manuscript is now according to your considerations.
Regards

Reviewer 2 Report
This manuscript has been revised by the authors as reviewers' comments. It has much improvement compared to the previous version.
Author Response
Dear reviewer, thank you for your comments
I`m attaching an improved version.
Regards.
